# Design of Protegrin-1 Analogs with Improved Antibacterial Selectivity

**DOI:** 10.3390/pharmaceutics15082047

**Published:** 2023-07-30

**Authors:** Ilia A. Bolosov, Pavel V. Panteleev, Sergei V. Sychev, Veronika A. Khokhlova, Victoria N. Safronova, Ilia Yu. Toropygin, Tatiana I. Kombarova, Olga V. Korobova, Eugenia S. Pereskokova, Alexander I. Borzilov, Tatiana V. Ovchinnikova, Sergey V. Balandin

**Affiliations:** 1M. M. Shemyakin & Yu. A. Ovchinnikov Institute of Bioorganic Chemistry, the Russian Academy of Sciences, 117997 Moscow, Russia; bolosov@ibch.ru (I.A.B.); p.v.panteleev@gmail.com (P.V.P.); ovch@ibch.ru (T.V.O.); 2V. N. Orekhovich Research Institute of Biomedical Chemistry, 119121 Moscow, Russia; 3State Research Center for Applied Microbiology & Biotechnology (SRCAMB), 142279 Obolensk, Russia; 4Department of Biotechnology, I. M. Sechenov First Moscow State Medical University, 119991 Moscow, Russia

**Keywords:** antimicrobial peptide, protegrin-1, oligomerization, cathelicidin, therapeutic index

## Abstract

Protegrin-1 (PG-1) is a cationic β-hairpin pore-forming antimicrobial peptide having a membranolytic mechanism of action. It possesses in vitro a potent antimicrobial activity against a panel of clinically relevant MDR ESKAPE pathogens. However, its extremely high hemolytic activity and cytotoxicity toward mammalian cells prevent the further development of the protegrin-based antibiotic for systemic administration. In this study, we rationally modulated the PG-1 charge and hydrophobicity by substituting selected residues in the central β-sheet region of PG-1 to design its analogs, which retain a high antimicrobial activity but have a reduced toxicity toward mammalian cells. In this work, eight PG-1 analogs with single amino acid substitutions and five analogs with double substitutions were obtained. These analogs were produced as thioredoxin fusions in *Escherichia coli*. It was shown that a significant reduction in hemolytic activity without any loss of antimicrobial activity could be achieved by a single amino acid substitution, V16R in the *C*-terminal β-strand, which is responsible for the PG-1 oligomerization. As the result, a selective analog with a ≥30-fold improved therapeutic index was obtained. FTIR spectroscopy analysis of analog, [V16R], revealed that the peptide is unable to form oligomeric structures in a membrane-mimicking environment, in contrast to wild-type PG-1. Analog [V16R] showed a reasonable efficacy in septicemia infection mice model as a systemic antibiotic and could be considered as a promising lead for further drug design.

## 1. Introduction

Host defense antimicrobial peptides (AMPs) are the key molecular factors of the innate immunity system of multicellular species, including humans [1]. Most AMPs have spatially separated hydrophobic and positively charged regions on their molecular surface, which facilitate their binding to negatively charged bacterial membranes and cell walls. Although AMPs tend to target ubiquitous and conserved molecular patterns associated with bacteria and fungi, many of them can also non-specifically damage host cell membranes due to their strong amphiphilic properties. In some cases, their selectivity can be improved by making minor modifications to the peptide structure using data on the structure–activity relationship (SAR) [2].

Among the best-studied vertebrate AMPs are cathelicidins—host defense peptides synthesized as precursor proteins containing a cathelin-like pro-domain [3]. Some cathelicidins are considered prototypes of a new generation of peptide antibiotics with immunomodulatory properties [4]. The group of cathelicidins includes peptides of various structural classes: α-helical linear, unstructured Pro- or Trp-rich, and β-hairpin Cys-containing peptides. The latter include a family of protegrins, 16–18 amino acid residues peptides, which were first isolated from porcine (*Sus scrofa*) leukocytes [5].

There are five naturally occurring protegrin isoforms reported to date [6]. All of them have a high homology, including four conserved Cys and several positively charged Arg residues. A key feature of protegrins as potential therapeutics is their compact structure, stabilized by two disulfide bonds (C6–C15 and C8–C13), which provides high stability for proteases [5,6,7,8]. The structure of protegrins resembles cyclic θ-defensins from rhesus macaque leukocytes [9] as well as tachyplesin and polyphemusin peptides found in horseshoe crab hemocytes [10]. Protegrins exhibit a broad spectrum of activity against Gram-positive and Gram-negative bacteria, including *E. coli*, *Pseudomonas aeruginosa*, and *Neisseria gonorrhoeae* [5,6,11], as well as against the yeast *Candida albicans* and the HIV-1 virus. The mechanism of the direct antimicrobial action of protegrins involves the formation of dimers from monomeric hairpins in the lipid bilayer environment, the self-association of the dimers into octameric β-barrels or tetrameric arcs, and, finally, pore formation leading to the uncontrolled transmembrane transport of ions and low-molecular-weight metabolites [12,13,14,15,16,17]. The complete disruption of the membrane by a detergent-like mechanism can be achieved at high concentrations of the peptide.

A number of studies focused on the development of modified protegrin analogs with improved medicinal properties. For example, there were attempts to obtain macrocyclic and cysteine-free analogs as well as analogs with replacement of selected amino acid residues [6,18,19]. Natural protegrins and their synthetic variants demonstrated considerable pharmacological potential. The best-known example is IB-367 (Iseganan), a truncated protegrin derivative, which passed phase III clinical trials as a topical antibiotic [20]. Nevertheless, advanced research on and the development of candidate protegrin analogs relying on recent fundamental studies of their mechanisms of action need a systemic approach. 

Previously, we conducted structural and functional studies to improve the therapeutical properties of a number of β-hairpin AMPs [21,22,23]. In this study, we performed a SAR study of protegrin-1 (PG-1) with the aim of mitigating its toxic side effects and designing novel therapeutically valuable analogs. The developed peptide variant demonstrated an improved selectivity in vitro and efficacy in a mouse infection model.

## 2. Materials and Methods

### 2.1. Production of Recombinant Protegrin Analogs

The recombinant plasmids for expression of AMPs were constructed using a pET-based vector (Novagen, Darmstadt, Germany), as previously described [23]. All protegrin analogs were obtained by heterologous expression in the *E. coli* BL21(DE3) (Novagen, Darmstadt, Germany) or ClearColi^®^ BL21(DE3) (endotoxin-free expression strain) (Research Corporation Technologies, Tucson, AZ, USA) system in a liquid LB medium, as previously described [24]. Protegrins were expressed as fusion proteins containing a modified thioredoxin A. Cell lysis, isolation of recombinant proteins by affinity chromatography, chemical cleavage, and purification of peptides and reverse-phase high-performance liquid chromatography (RP-HPLC) were performed, as previously described [24]. Ni^2+^ Sepharose (Cytiva Life Sciences, Marlborough, MA, USA) and ReproSil-Pur 120 C18-AQ, 5 µm column (Dr. Maisch GmbH, Ammerbuch-Entringen, Germany) were used for affinity chromatography and RP-HPLC, respectively. The purified peptides were collected and analyzed by MALDI-TOF MS using a Reflex III mass-spectrometer and flexAnalysis 3 software (Bruker Daltonics GmbH & Co. KG, Bremen, Germany). 

### 2.2. Antimicrobial Assay

The antimicrobial activity of the peptides was determined using two-fold serial dilutions in sterile 96-well flat-bottom polystyrene microplates (Eppendorf, Hamburg, Germany) (Cat. No. 0030730011), as previously described [24]. The following strains of bacteria were used: *E. coli* ATCC 25922, *E. coli* ML35p (ATCC collection), *E. coli* BW 25113 (Keio collection), *Pseudomonas aeruginosa* PAO1 (ATCC collection), *P. aeruginosa* ATCC 27853, *Klebsiella pneumoniae* ATCC 700603, *Staphylococcus aureus* 209P (ATCC collection), *S. aureus* ATCC 29213, and *Micrococcus luteus* B-1314 (VKM collection). *Acinetobacter baumannii* (strain XDR CI 2675) was provided by the I.M. Sechenov First Moscow State Medical University hospital. After a 24 h incubation period, the metabolic activity of the bacteria was assessed by adding 20 µL of a redox indicator, resazurin (Sigma-Aldrich, St. Louis, MO, USA), at a concentration of 0.1 mg/mL, to the wells. The plate was then incubated for a further two hours. The minimum inhibitory concentration (MIC) was determined as the lowest concentration of the peptide at which no color change in the resazurin indicator was observed. The results were presented as median values, which were determined based on at least three independent experiments carried out in triplicate.

### 2.3. Hemolysis and Cytotoxicity Assay

To assess the hemolytic activity of the compounds, freshly isolated human erythrocytes (hRBCs) were used according to the methodology described in a previous study [22]. Melittin was used as a positive peptide control that induces significant hemolysis. Isotonic phosphate buffer (PBS) served as a negative control, while 0.1% Triton X-100 solution was used as a control for complete hemolysis.

In order to evaluate the cytotoxicity of the peptides against a human keratinocyte cell line (HaCaT, Cat. No. 300493) (Cell Lines Service, Köln, Germany), the colorimetric 3-(4,5-dimethylthiazol-2-yl)-2,5-diphenyltetrazolium bromide (MTT) dye reduction assay was performed, as described in a previous publication [23]. Wells containing no peptide were used as a 100% survival control. Each peptide was tested in two separate experiments to ensure the reliability of the results.

### 2.4. Bacterial Membranes’ Permeability Assay

The efficacy of the peptides in permeabilizing bacterial membranes was assessed using a previously established method [24]. The assay utilized o-nitrophenyl-β-D-galactopyranoside (ONPG, AppliChem GmbH, Darmstadt, Germany) and the *E. coli* strain ML35p. Control experiments were conducted under identical conditions but without the addition of the peptide. To monitor the progress of the assay, the optical density of the solution was measured at regular intervals. Measurements were taken every 5 min for the first 90 min, followed by measurements at 100, 110, 120, 150, and 180 min. To ensure reliability, three independent experiments were performed, and it was determined that the observed curve pattern was consistent across all three series of experiments. The results show plots from one of these series.

### 2.5. CD and FTIR Spectroscopy

Peptides PG-1 and [V16R] were subjected to Fourier-transform infrared (FTIR) spectroscopy and circular dichroism (CD) spectroscopy measurements in different solvents, water, EtOH, and dodecylphosphocholine (DPC) micelles, following the methodology outlined in previous studies [25,26]. The concentration of the peptides used in the measurements was 200 µM. For measurements in water or EtOH, the peptide samples were dissolved in the appropriate medium. For measurements in the presence of DPC micelles, an aqueous solution of DPC (Avanti Polar Lipids, Alabaster, AL, USA) was added to the peptide sample dissolved in water, resulting in a final peptide:lipid ratio of 1:150. The obtained spectroscopic data were analyzed and fitted using OriginPro 8.5 software (OriginLab Corp., Northampton, MA, USA).

### 2.6. Animal Studies

Experiments were performed with male and female 8–10-week-old BALB/c mice (22–24 g, “Andreevka” laboratory animal nursery FMBA, Russia) according to the previously described method [27]. Briefly, a total of 3 groups of 5 mice per group were infected by intraperitoneal (i.p.) injection of the bacterial inoculum (*E. coli* ATCC 25922), 10^6^ colony forming units (CFU) per animal, in the presence of mucin. All the animals received two i.p. injections: 1 h and 4 h after bacterial challenge. The first group received ciprofloxacin (Sigma-Aldrich, St. Louis, MO, USA) as a positive control at a dose of 10 mg/kg per injection. The second group received peptide [V16R] at a dose of 5 mg/kg per injection. The third group received PBS as a vehicle control. Animals were monitored for 7 days post-infection. All the surviving mice were euthanized by CO_2_ asphyxiation. The spleen was aseptically removed, homogenized, serially diluted, and plated on Endo agar to detect CFU in a sample.

All animal experiments were performed at the State Research Center for Applied Microbiology and Biotechnology (SRCAMB) in Obolensk (Russia). Experiments were approved by the Institutional Bioethics Committee of the SRCAMB and performed according to the Russian Federation’s rules and Directive 2010/63/EU of the European Parliament and of the Council.

## 3. Results and Discussions

### 3.1. Peptide Design and Expression and Purification of Protegrin Analogs

The extended structure–activity relationship studies previously conducted for PG-1 led to the design of IB-367 (also known as Iseganan); however, it did not exhibit higher levels of activity than the natural peptide and was also toxic to normal mammalian cells [20]. Although several low-toxic PG-1 analogs were developed as the result of SAR studies, the molecular basis for such a decrease in cytotoxicity in vitro has not yet been understood. In this work, we used the design strategy based on our previous studies of different β-hairpin AMPs; we aimed to reduce the overall hydrophobicity, oligomerization, and aggregation tendency while retaining the β-hairpin structure. The disulfide bridges were necessary to maintain the high stability and biological activity of the β-hairpin peptides, which is why we kept all the cysteine residues, even though they contribute to the overall hydrophobicity. Previously, we showed, for arenicin-1, another β-hairpin AMP, that the formation of a highly amphipathic dimeric structure predetermines its activity against eukaryotic membranes and promotes cytotoxicity [21]. Its mutant analog, Ar-1 [V8R], which shows similar antibacterial activity, has a significantly lower dimerization propensity in the membrane-mimicking environment of DPC micelles and reduced hemolytic and cytotoxic activities. Therefore, we decided to apply a similar approach to optimize the structure of PG-1, with activity against membranes built from zwitterionic lipids that was also attributed to its aggregation tendency [28]. We assumed that the introduction of a positive charge at certain positions of the β-sheet region would affect the ability of the peptide to dimerize (or to form oligomeric structures) and aggregate. In turn, this may change its biological activity. For this purpose, we obtained a number of analogs of natural PG-1 with hydrophobic residues substituted by arginine, namely, [V14R], [V16R], [Y7R], and [L5R]. The design of the experiment is presented in Figure 1A,B.

In addition to the possible electrostatic repulsion between the monomers, the introduction of arginine in the β-sheet region can also stabilize the interactions between the peptide and negatively charged bacterial lipids through the formation of guanidine–phosphate complexes [29,30,31]. For a more gradual reduction in hydrophobicity, a number of analogs were obtained by replacing leucine and valine with less hydrophobic alanine: [L5A], [V14A], and [V16A]. Additionally, an analog with [Y7T] substitution was designed to reduce the hydrophobicity of the side chain, while keeping the hydroxyl group in its structure. After receiving primary data on the antimicrobial activity of the first series of analogs, a set of double mutants was obtained as well (Figure 1B). 

Natural cathelicidins do not undergo significant post-translational modifications (except for a *C*-terminal amidation and disulfide bond formation), so we chose heterologous expression in *E. coli* as the method for their production. This allowed us to avoid focusing on the design of analogs with the cheaper technology of chemical synthesis (one of the main criteria that guided the development of IB-367). Previously, recombinant expression in *E. coli* was used to produce protegrin via green fluorescent protein (GFP) fusion [32]. In this study, the modified thioredoxin A was used as a fusion partner that increased the solubility of the recombinant protein, promoted the correct disulfide bond formation, and masked the toxic effects of AMPs. For all the obtained analogs, the final yield of a pure peptide ranged from 4.0 to 7.2 mg per 1 L of bacterial culture (Appendix A). The experimentally measured m/z values of the recombinant peptides were in good agreement with the corresponding calculated molecular masses (Figure 1B).

### 3.2. In Vitro Biological Assays

The therapeutic potential of the obtained protegrin analogs was evaluated based on the selectivity of their action against bacterial cells compared to human cells—erythrocytes and HaCaT. The antimicrobial activity of the analogs was determined against a panel of Gram-negative and Gram-positive bacteria using the two-fold serial dilution method (Table 1). Iseganan was used as a control.

None of the mutant analogs showed a significant increase in antimicrobial activity compared to the natural PG-1, although in some cases the MIC values against certain strains were lower by one dilution step. Three alanine mutants, [L5A], [V14A], and [V16A], showed the highest activity, close to that of the original peptide. In particular, only these three analogs retained baseline activity against *P. aeruginosa* and *K. pneumoniae* strains. In the case of different strains of *E. coli*, the activity of most analogs, with the exception of [V14R], [Y7T,V16R], [V14A,V16R], and [L5A,V16R], was comparable to the activity of the natural peptide. Importantly, there was a major difference in the activity of arginine mutants V14R and V16R.

In order to investigate the effect of these mutations on the interaction of peptides with the eukaryotic membrane, a hemolytic test was performed against freshly isolated human erythrocytes (Figure 2A). All mutants showed reduced activity compared to PG-1 and Iseganan. As expected, the most pronounced decrease was observed for the arginine mutants. The hemolytic test data were confirmed using the MTT assay for HaCaT cells after 24 h incubation (Figure 2B,C). There was a good correlation between a decrease in the level of hemolytic activity and an increase in the survival of the test culture. The observed level of cell viability in the presence of Iseganan was surprisingly high, apparently due to its effective binding to serum proteins. In addition, analogs [L5A,V16A] and [Y7T] were more toxic to normal cells than would be expected based on their hemolytic properties. In both assays, the [V14R] mutant was less active than [V16R].

In order to estimate the possible therapeutic window for the studied analogs, a therapeutic index was calculated (Table 2). Analog [V16R] demonstrated the highest therapeutic index: it was significantly less toxic against normal cells than the natural peptide, retaining almost the same level of antimicrobial activity.

To study the mechanism of action of the peptides, we tested their ability to disrupt the integrity of *E. coli* ML35p membranes using a chromogenic substrate, ONPG, for the cytoplasmic β-galactosidase (Figure 3). PG-1 showed a strong membranolytic effect in a concentration range corresponding to its MIC values (Figure 3A), in good agreement with the literature data on the main mechanism of action of this peptide [12,13,14,15,16,17].

The most selective analog, [V16R], showed a somewhat different pattern compared to PG-1 (Figure 3B). Although the MICs against *E. coli* ML35p were equal for both peptides (0.25 μM), analog [V16R] at this concentration had almost no effect on ONPG permeation, in sharp contrast to the wild-type peptide. This indicates a change in the mechanism of action of the mutant peptide. We assume that this amino acid substitution may prevent the peptide oligomerization during its interaction with the lipid bilayer, affecting the pore size and/or specificity.

For analog [V14R], the curve shape changed even more (Figure 3C). Both [V14R] and [V16R] demonstrated slow kinetics of membrane permeability, similar to the effect of capitellacin, another β-hairpin AMP, as previously observed [33]. At first glance, the valine residues at these positions should play a similar role in PG-1 dimers’ formation (Figure 1C,D). However, all our in vitro assays indicated that the introduction of a positively charged group in position 14 has a much stronger destabilizing effect. Apparently, due to the more central position of the V14 residue, the introduction of a positive charge leads to a strong electrostatic repulsion and prevents the formation of dimers.

The three alanine mutants we obtained, [L5A], [V14A], and [V16A], showed average MIC values (geometric mean of 0.4 µM; Table 2) identical to PG-1. We suggest that this type of substitution does not affect the dimerization process and the mechanism of action of these peptides but makes them less toxic by reducing the overall hydrophobicity.

All analogs with double amino acid substitutions (especially those with arginine) showed a good reduction in toxic properties. However, at the same time, the activity against bacterial strains significantly decreased (four–eight fold). This seems to be due to a decrease in the overall hydrophobicity of the peptide below the critical level. Despite the encouraging values of the therapeutic index, these analogs lose their practical significance due to the need for higher doses of the drug.

### 3.3. Spectroscopy Analysis of PG-1 and Its Analog [V16R]

The dimerization and subsequent oligomerization of protegrins was previously shown to play an important role in their membranotropic activity [12]. Various models assume the formation of a parallel [12,14,15,34] or antiparallel [35] dimer, followed by the assembly of larger complexes, in particular an octamer structure [17,34]. We hypothesized that the formation of an oligomeric complex plays an important role in the process of the disruption of the eukaryotic membrane. To test this hypothesis, studies of natural PG-1 and its analog with reduced toxicity, [V16R], were carried out using CD and FTIR spectroscopy.

The CD spectra of the peptides in water, ethanol, and DPC micelles are shown in Figure 4. For both peptides, a spectrum in ethanol has a minimum at 206 nm (n → π* amide transition) and a maximum near 190 nm (π → π* amide transition), which is typical for β-stranded polypeptides [36,37,38] and β-hairpins. Both spectra in water suggest the presence of a flexible chain [36,37,38], while the increase in molar ellipticity of the band at ~190 nm and a small red shift from 200–204 nm to 208 nm in ethanol (more pronounced for the [V16R] analog) indicate a more ordered structure in a nonpolar environment. A similar but less pronounced pattern was observed for DPC micelles. Thus, both peptides have the characteristic spectrum of a β-hairpin structure in all the tested environments.

While the CD spectra of PG-1 and its analogs only indicate the formation of a β-structure, FTIR spectroscopy may provide a more precise picture of a peptide structure. The detection of the different secondary structural elements by FTIR spectroscopy is fundamentally related to the analysis of hydrogen bonds (H-bonds). Here, we analyzed the bands of PG-1 and [V16R] in the amide I region of FTIR spectra, which are mainly associated with the C = O stretching vibration (Figure 5). Both peptides have similar spectra in EtOH, with two major bands at ~1677 and ~1644 cm^−1^ (Figure 5B), which are typical for a peptide forming a β-sheet structure [39]. The strong band at 1676 cm^−1^, which is also dominant for both peptides in water (Figure 5A), corresponds to weakly solvated amide carbonyls that are not involved in intramolecular H-bonds [40,41], as we discussed for the β-hairpin antimicrobial peptides arenicin-2 [25] and dodecapeptide [26]. The band at ~1644 cm^−1^ corresponds to hydrogen-bonded C = O groups. This band correlates with the calculated FTIR spectra for the mixed parallel/antiparallel β-strands [39,42]. Thus, the shape of the spectra in ethanol suggests the formation of peptide dimers.

The strongest difference between the FTIR spectra of PG-1 and [V16R] is observed in the presence of DPC (Figure 5C). The predominance of the 1643 cm^−1^ component in the case of PG-1 indicates the presence of a β-structure with a greater number of hydrogen-bonded groups than in the dimer [42]. The absence of a 1677 cm^−1^ band for PG-1 indicates a low portion of carbonyls that are not involved in H-bonds. Therefore, the formation of an octamer structure with the compact packing of strands can be assumed in DPC micelles, as was shown for PG-5 [17]. In contrast, the FTIR spectrum shape of [V16R] in DPC, which is similar to that in EtOH, is more consistent with a dimer structure. This confirms the previous suggestion that the introduction of an additional charged arginine residue in the β-sheet region of PG-1 prevents the dense packing of β-strands and further oligomerization. The loss of the ability to form oligomer-based compact pores may explain the modest membranotropic activity and slow membrane permeabilization of analog [V16R] compared to PG-1 (Figure 3). At the same time, the question of a possible dimer topology for analog [V16R] remains open.

### 3.4. In Vivo Assay

Peptide [V16R], which demonstrated the highest therapeutic index among the tested PG-1 analogs, was selected as a lead candidate for study in a mouse model of bacterial infection. For animal experiments, the recombinant peptide was produced in an endotoxin-free ClearColi^®^ BL21(DE3) expression system. The trifluoroacetate (TFA) counterions were replaced with chloride ions by an incubation of the peptide with 5 mM HCl followed by lyophilization, according to [43]. The efficacy of the peptide in a systemic septicemia infection mice model was examined using five animals per group (Figure 6). Due to the lack of information on the acute toxicity and pharmacokinetics of the PG-1 analog, a relatively low dose of the peptide (5 mg/kg) was used in the experiment. The antibiotic ciprofloxacin (10 mg/kg) was used as a control. Both were administered twice within the first day after infection. Intraperitoneal infection of BALB/c mice with a suspension of *E. coli* ATCC 25922 in the presence of mucin resulted in death within two days for four out of five mice treated with the vehicle control (PBS) and the survival of all mice treated with ciprofloxacin. The high CFU burden was confirmed in mice from the negative control group. 

Here, we demonstrated a therapeutic efficacy of 60% (the death of two out of five mice) when applying a double dose of the PG-1 analog [V16R] after a one-week experiment. As expected, we did not identify *E. coli* in the spleen after euthanizing all the surviving animals. These preliminary in vivo data suggest that modified recombinant variants of β-hairpin AMPs are promising anti-Gram-negative antimicrobials. However, some optimizations can be made to further improve the efficacy of [V16R]. First, a prolonged therapy with increased doses may be tested after a study of tolerability in mice and a maximum tolerated dose (MTD) evaluation. Second, further structure modifications can be made: in particular, the acetylation and/or amidation of the peptide termini to improve its resistance to serum proteases.

The ability of PG-1 to protect mice against a lethal bacterial challenge was previously demonstrated: for peritoneal infections with *S. aureus* or *P. aeruginosa*, PG-1 administered i.p. at a dose of 0.5 mg/kg reduced mortality from 93–100% in the vehicle control group to 0–27% [44]. In these studies, PG-1 was administered to immunocompetent animals immediately after the bacterial challenge. In our work, peptide [V16R] was administered to mucin-treated mice 1 h after the bacterial challenge, which could also reduce the protective effect.

## 4. Conclusions

Despite the growing number of SAR studies on β-hairpin AMPs and the development of powerful computational approaches, principles for the prediction of the biological activities of these molecules are still obscure. A number of SAR studies on protegrins were carried out earlier, determining the contributions of a peptide’s length, charge, and hydrophobicity as well as the role of disulfide bonds [6,45,46]. The main identified patterns can be summarized as follows: (i) analogs with reduced positive charge are less active; (ii) there are no stereospecific interactions between PG-1 and bacterial cells; (iii) the overall structure (amphipathicity, charge, and stabilized β-hairpin fold) is more important for activity than the presence of specific amino acid residues; (iv) the introduction of polar residues into the hydrophobic β-sheet region generally reduces any activity. 

Our work was aimed at filling the gap between the SAR analysis of the β-sheet region of protegrins and studies devoted to the problem of oligomerization and selectivity of action of these peptides. Here, we showed that

A slight decrease in the hydrophobicity of a PG-1 β-sheet region (for example, [L5A], [V14A], or [V16A]) may improve peptide selectivity without increasing the MIC values against bacteria.Double amino acid substitutions in the β-sheet region of PG-1 reduce both antibacterial activity (to a lesser extent toward Gram-negative bacteria) and cytotoxicity ≥10-fold.The amino acid residue at position 14 is important for biological activity: analog [V14R] has a 20-fold lower antibacterial activity compared to wild-type PG-1.The arginine mutants [Y7R], [V14R], and [V16R] (but not [L5R]) have significantly reduced hemolytic activities. Analysis of analog [V16R] by FTIR spectroscopy showed that the peptide is unable to form oligomeric structures in eukaryotic membrane-mimicking environment, in contrast to wild-type PG-1.The decrease in the antibacterial activities of the tested PG-1 analogs generally correlates with a decrease in hemolytic activity and cytotoxicity. The only exception found is analog [V16R].

Peptide [V16R] is the most selective analog, with a TI of 60.2 (compared to 1.8 for PG-1), which has a reasonable efficacy in vivo as a systemic antibiotic and can be considered as a promising lead for further drug design.

## Figures and Tables

**Figure 1 pharmaceutics-15-02047-f001:**
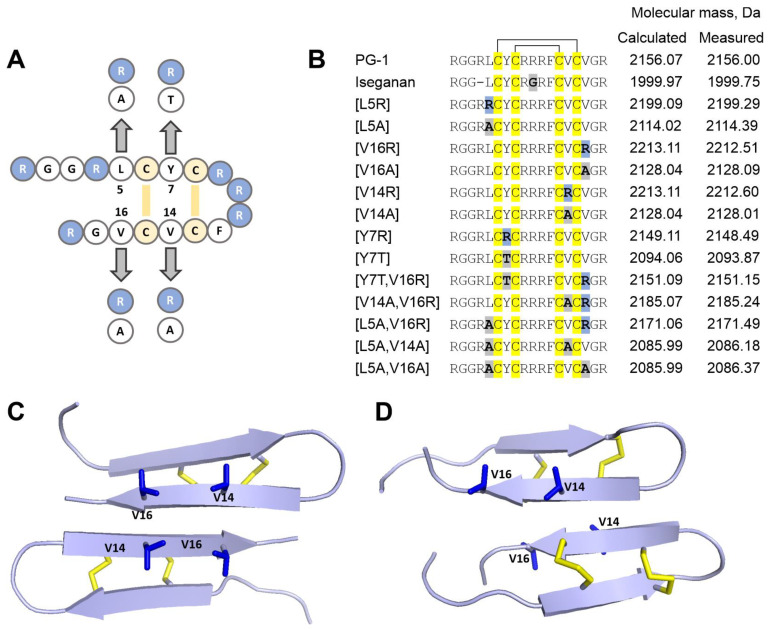
Amino acid replacements in the structure of PG-1 (**A**) and the list of obtained analogs of protegrin (**B**). Spatial structures of antiparallel (**C**) (protegrin-3, PDB ID 2MZ6) and parallel (**D**) (protegrin-1, PDB ID 1ZY6) dimers of protegrin in eukaryotic membrane-mimicking environment.

**Figure 2 pharmaceutics-15-02047-f002:**
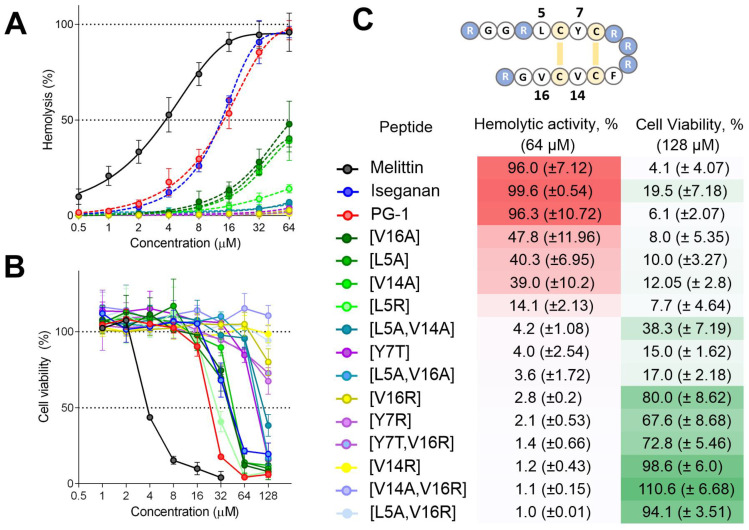
(**A**) Hemolytic activity of the peptides after 1.5 h incubation (hemoglobin release assay) and (**B**) cytotoxicity against human keratinocyte cell line (HaCaT) after 18 h incubation (MTT assay). The data are presented as the mean ± SD of two independent experiments. (**C**) Comparison of hemolytic activity values and HaCaT live cells (MTT assay). The background is colored using a heat map reflecting the degree of hemolysis (red) and cell viability (green).

**Figure 3 pharmaceutics-15-02047-f003:**
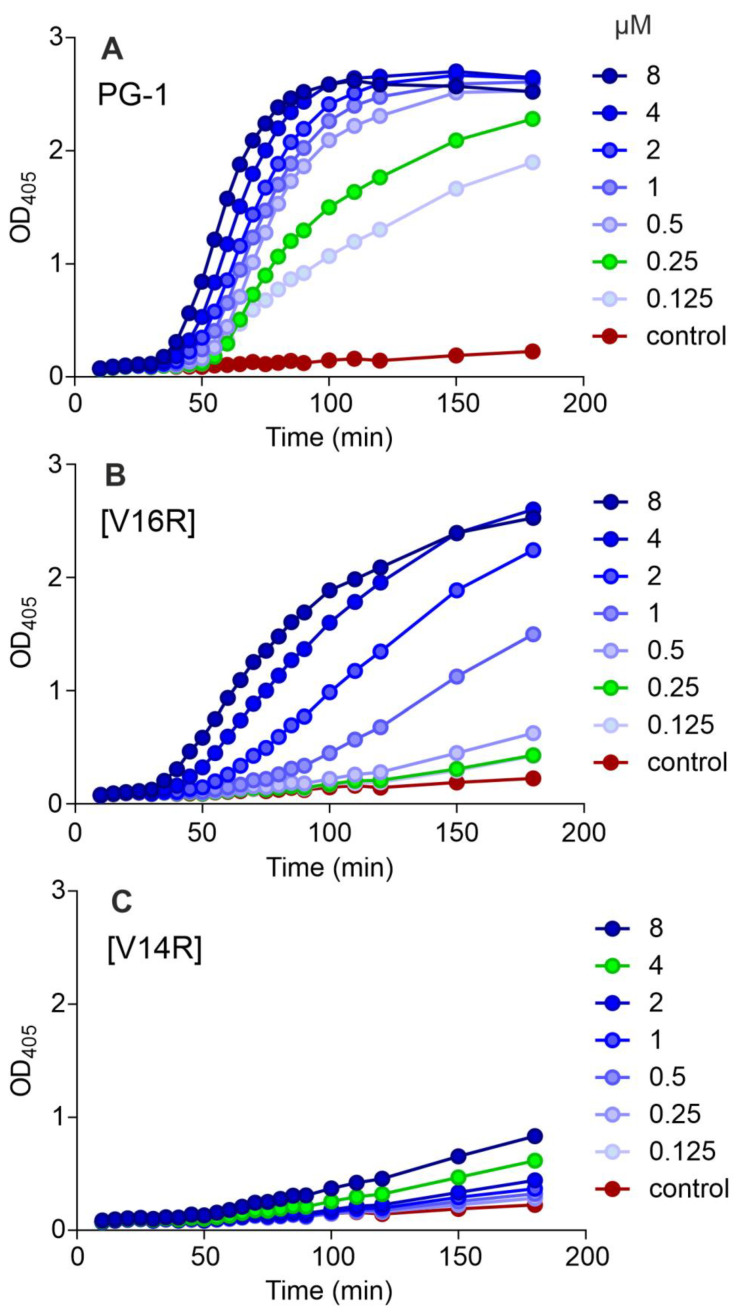
The effect of PG-1 and its analogs at different concentrations on the permeability of the membranes of *E. coli* ML-35p (ONPG assay). The concentration corresponding to the MIC value against *E. coli* ML-35p is shown in green.

**Figure 4 pharmaceutics-15-02047-f004:**
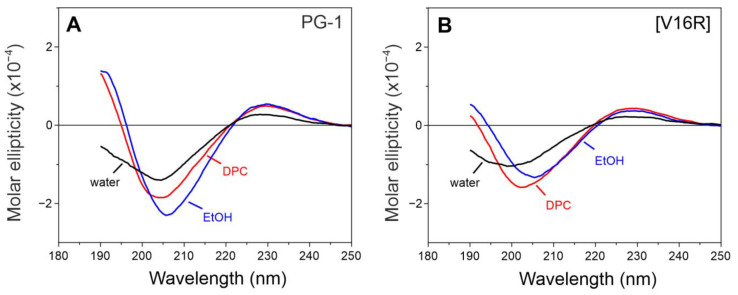
CD spectra of PG-1 (**A**) and [V16R] (**B**) in different environments. Spectra in water (black), ethanol (blue), and DPC (red).

**Figure 5 pharmaceutics-15-02047-f005:**
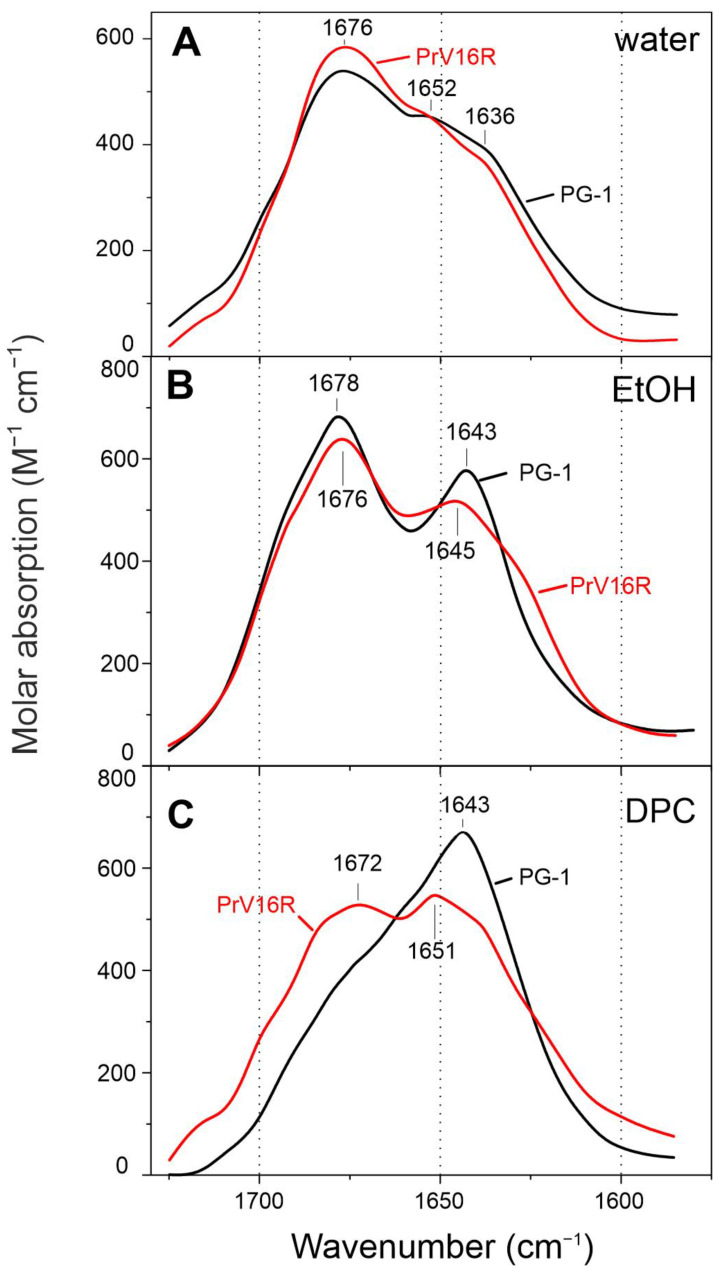
FTIR spectra of PG-1 (black) and [V16R] (red) in water (**A**), ethanol (**B**), and DPC (**C**).

**Figure 6 pharmaceutics-15-02047-f006:**
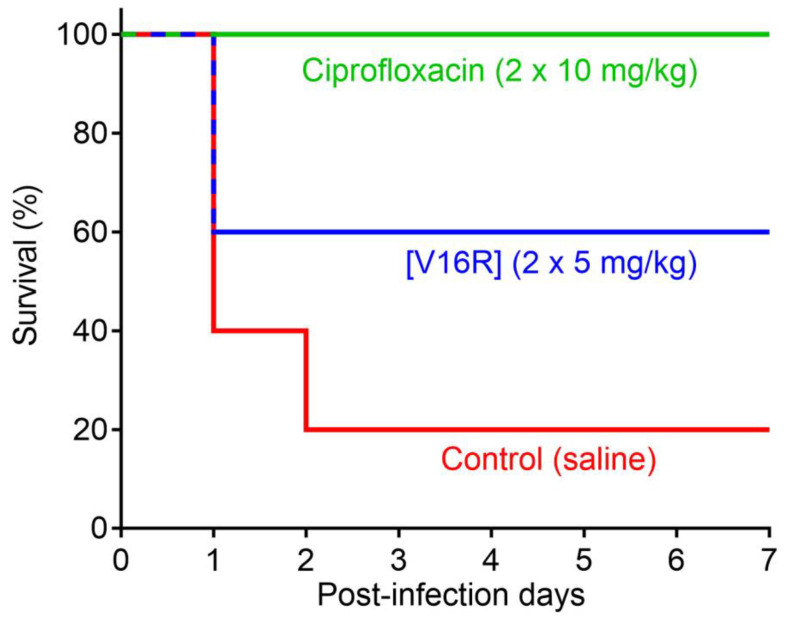
In vivo efficacy study of the selected protegrin-1 analog [V16R]: survival rates of BALB/c mice (n = 5) intraperitoneally infected with *E. coli* ATCC 25922 (10^6^ bacteria in the presence of 2.5% mucin). Ciprofloxacin (two injections of 10 mg/kg) and saline were used as positive and negative control, respectively. The health status of the mice was checked once a day for 7 days after infection.

**Table 1 pharmaceutics-15-02047-t001:** Antimicrobial activity of protegrin-1 and its analogs.

Peptide	MIC *, μM									
	*E. coli* ATCC 25922	*E. coli* ML35p	*E. coli* BW 25113	*P. aeruginosa* PAO1	*P. aeruginosa* ATCC 27853	*A. baumannii*	*K. pneumoniae* ATCC 700603	*S. aureus* 209P	*S. aureus* ATCC 29213	*M. luteus* B-1314
PG-1	0.5	0.25	0.125	0.5	0.25	0.25	1	0.5	1	0.25
Iseganan	0.5	0.5	0.5	8	4	0.5	4	2	4	0.5
[L5R]	1	0.5	0.25	1	1	0.5	8	4	4	1
[L5A]	0.5	0.25	0.125	0.5	0.25	0.25	1	1	1	0.5
[V16R]	0.5	0.25	0.25	1	0.5	0.125	4	4	4	1
[V16A]	0.5	0.125	0.25	0.25	0.5	0.125	1	1	1	0.5
[V14R]	4	4	2	32	>32	2	32	32	32	2
[V14A]	0.5	0.25	0.125	0.5	0.25	0.125	1	1	1	0.5
[Y7R]	0.5	0.5	0.5	8	4	0.5	8	16	16	2
[Y7T]	1	0.5	0.5	2	2	0.5	8	16	16	4
[Y7T,V16R]	8	4	8	>32	>32	4	>32	>32	>32	4
[V14A,V16R]	2	1	2	8	4	0.5	32	32	32	4
[L5A,V16R]	2	1	2	8	4	0.5	32	32	32	4
[L5A,V14A]	0.5	0.5	0.5	1	1	0.25	8	4	>32	4
[L5A,V16A]	0.5	0.5	0.5	1	1	0.25	8	4	>32	4

* Median MIC value for three independent experiments.

**Table 2 pharmaceutics-15-02047-t002:** Biological activity of protegrin analogs and their therapeutic indices.

	MHC ^a^	MIC (GM) ^b^	TI ^c^
PG-1	0.7	0.4	1.8
Iseganan	1	1.4	0.7
[L5R]	4.1	1.2	3.3
[L5A]	4.1	0.4	9.4
[V16R]	49.2	0.8	60.6
[V16A]	4.3	0.4	10.6
[V14R]	>64	9.8	13.0
[V14A]	5.5	0.4	13.5
[Y7R]	61.4	2.5	24.9
[Y7T]	33.5	4.4	7.6
[Y7T,V16R]	>64	21.1	6.1
[V14A,V16R]	>64	4.9	26.0
[L5A,V16R]	>64	4.9	26.0
[L5A,V14A]	18.8	4.5	4.2
[L5A,V16A]	11.9	4.5	2.7

^a^ The MHC was determined as the minimum hemolytic concentration that caused 2% hemolysis of fresh human red blood cells (hRBCs). The value of 128 μM was used for the calculation, when less than 2% hemolysis was achieved at a concentration of 64 μM. ^b^ The geometric mean of the peptide’s MICs (GM) against all the test strains. ^c^ The therapeutic index (TI) is the ratio of the MHC to the MIC (GM). Larger values indicate greater peptide selectivity.

## Data Availability

Not applicable.

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
