# Peer review of "Design of Protegrin-1 Analogs with Improved Antibacterial Selectivity"

_pharmaceutics, 2023, doi:10.3390/pharmaceutics15082047_

Round 1

Reviewer 1 Report

The article is valuable and well structured. However, it needs a minor revision. It should be supplemented with expansions of some abbreviations (e.g. pET). It happens that the expansion is not the first time the abbreviation is used in the text, but the next time (e.g. TFA). This should be sorted out. In addition, there is no information about the origin of the bacterial strains used in Production of Recombinant Protegrin Analogs, Antimicrobial Assay and Bacterial Membranes Permeability Assay. The same applies to the HaCaT cell line towards which cytotoxicity was tested.

The English language used is correct but you should correct typos.

Reviewer 2 Report

The manuscript showed interesting results and is potential for publication.

Some minor corrections are needed

Reviewer 3 Report

The manuscript entitled “Design of protegrin-1 analogs with improved antibacterial selectivity” (pharmaceutics-2521557) provides insights into the development of mutant protegrin-1 (PG-1) analogs for improved antibacterial selectivity and decreased hemolytic activity. The goal was to synthesize derivatives by maintaining the β-hairpin structure but with reduced hydrophobicity, aggregation, and oligomerization tendency. The comparison between analogs V14R and V16R for permeability describes the potential difference in activity due to positional change in the amino acids. The study was rationally designed, and the results were well presented.

However, the following concerns need to be addressed before accepting the manuscript for publication.

1.      Why didn’t the authors discuss the CD spectra in DPC (but represented in Figure 4) and mentioned twice about CD spectra in ethanol? Is this a typographical error or does it not contribute to the study?

2.      The authors compared the selectivity of PG-1 analogs by comparing the activity against bacterial and human cells. Is there any specific reason behind choosing HaCaT cells?

3.       The reference for the in vitro cytotoxicity assay using MTT should be included.

4.      Minor typographical errors like spacings should be checked for in the entire manuscript.

5.      The uniformity in the references should be maintained. For example,

a.       In references 6, 20, 28, 30: All the author names should be provided.

b.      In references 13, 30, 31: The capitalization of all the species names in the title should be corrected.

c.       Reference 21, 41: Journal’s name should be abbreviated.

d.      Reference 24, 41: Page numbers should be included.

Reviewer 4 Report

comments for pharmaceutics-2521557

The peptide designed in this article has the potential to solve some of the possible clinical clinically. But there are still some problems in the article.

1. In the abstract, fewer results are described.

2. In the preface, the purpose of the trial expression is not very clear.

3. Figure S2 Should be placed in the main text.

4. Data have no error lines in Figure 3.

5. In the Supplementary Material, the SDS-PAGE plots after expression should be added.

6. Each peptide should display plots as in C and D in Figure 1.

7. In Figure 6, the data on days 0-1 feel somewhat problematic and are unclear about the number of mice in each group.

8. In mouse trials, the results of PG-1 should be added.

9. The animal test section says that the mouse spleen was collected, but there are no data in the results.

10. The conclusion does not match the test purpose.
